# Liver Immune Microenvironment and Metastasis from Colorectal Cancer-Pathogenesis and Therapeutic Perspectives

**DOI:** 10.3390/cancers13102418

**Published:** 2021-05-17

**Authors:** Xuezhen Zeng, Simon E. Ward, Jingying Zhou, Alfred S. L. Cheng

**Affiliations:** 1Department of Liver Surgery, The First Affiliated Hospital, Sun Yat-sen University, Guangzhou 510080, China; zengxzh7@mail.sysu.edu.cn; 2Department of Pharmacy, The First Affiliated Hospital, Sun Yat-sen University, Guangzhou 510080, China; 3Institute of Precision Medicine, The First Affiliated Hospital, Sun Yat-sen University, Guangzhou 510080, China; 4Medicines Discovery Institute, Cardiff University, Cardiff CF10 3AT, UK; WardS10@cardiff.ac.uk; 5School of Biomedical Sciences, The Chinese University of Hong Kong, Hong Kong 999077, China

**Keywords:** liver immune microenvironment, colorectal cancer liver metastasis, therapeutic perspectives

## Abstract

**Simple Summary:**

Liver metastasis remains the major contributor in colorectal cancer-related death. It has become clear that the unique immune features of liver microenvironment take part in many steps of metastatic cascade, from pre-metastatic niche formation, tumor cell colonization to metastatic tumor establishment. Therefore, better understanding of mechanisms orchestrating the formation of a hospitable hepatic metastatic niche is necessary for the development of effective therapies. This review summarizes the current understandings of the critical role of liver immune microenvironment in metastasis development and provides therapeutic perspective on targeting the metastasis-prone microenvironment.

**Abstract:**

A drastic difference exists between the 5-year survival rates of colorectal cancer patients with localized cancer and distal organ metastasis. The liver is the most favorable organ for cancer metastases from the colorectum. Beyond the liver-colon anatomic relationship, emerging evidence highlights the impact of liver immune microenvironment on colorectal liver metastasis. Prior to cancer cell dissemination, hepatocytes secrete multiple factors to recruit or activate immune cells and stromal cells in the liver to form a favorable premetastatic niche. The liver-resident cells including Kupffer cells, hepatic stellate cells, and liver-sinusoidal endothelial cells are co-opted by the recruited cells, such as myeloid-derived suppressor cells and tumor-associated macrophages, to establish an immunosuppressive liver microenvironment suitable for tumor cell colonization and outgrowth. Current treatments including radical surgery, systemic therapy, and localized therapy have only achieved good clinical outcomes in a minority of colorectal cancer patients with liver metastasis, which is further hampered by high recurrence rate. Better understanding of the mechanisms governing the metastasis-prone liver immune microenvironment should open new immuno-oncology avenues for liver metastasis intervention.

## 1. Introduction on Colorectal Cancer (CRC) Liver Metastasis

### 1.1. Liver Tropism in Cancer Metastasis

Cancer metastasis is the major obstacle to successful management of malignant disease and accounts for approximately 90% of cancer related mortality [1]. Interestingly, the formation of metastasis favors a few target organs, including liver, bone marrow, etc. Liver is one of the most common metastatic sites for various malignancies [2], including colorectal cancer, pancreatic cancer, gastric cancer, breast cancer, and melanoma, etc. Metastatic tumor cells usually invade into the liver parenchyma for seeding and progress. However, the mechanisms underlying liver tropism in cancer metastasis remain largely unknown.

Numerous studies addressed that the unique structure and characteristic of liver that it enriches in vessels with high permeability and has unparalleled dual blood connectivity, and the immunosuppressive microenvironment, make it vulnerable to the seeding of disseminated tumor cells [3]: The dual vascular supply of the liver by the systemic arterial and portal venous system enhances the entrapment of circulating tumor cells, explaining increased incidence of liver metastasis in patients with gastrointestinal cancers; In addition, the immune microenvironment in liver has evolved to dampen immunity to neoantigens entering the liver from the gut via portal vein so as to avoid damage to the liver [4].

The liver is comprised of heterogeneous cell populations, including parenchymal hepatocytes, and nonparenchymal cells like hepatic stellate cells, infiltrated or resident immune cells, and liver sinusoidal endothelial cells. Accumulating evidence shows that both the parenchymal and the nonparenchymal cells play a role in the process of metastatic cascade, including facilitating acquisition of epithelial–mesenchymal transition (EMT) phenotype, migration to liver, seeding, and colonization as well as the decision to undergo dormancy versus outgrowth [3].

### 1.2. Clinical Significance of CRC Liver Metastasis

Among all types of cancer, CRC is the most common cancer that predominantly metastasizes to the liver [5]. CRC is the fourth most common and third most deadly malignancy worldwide with a steadily rising incidence rate in developing countries [6]. Approximately 50% of CRC patients have already developed liver metastases at diagnosis, and 40–50% of patients will develop liver metastasis after primary tumor resection within 3 years [7,8,9]. Consequently, emergence of liver metastasis has been used as prognostic marker for CRC. The 5-year survival rate was dramatically decreased to 10–20% compared to 80–90% of patients with only localized CRC [10].

CRC liver metastases mainly exhibit two distinct histopathological growth patterns (HGPs) at the interface between the tumor and surrounding liver parenchyma, namely, desmoplastic type (dHGP) and non-desmoplastic type [11]. dHGP is characterized by increased angiogenesis, and the new blood vessels appear leaky and are functionally impaired with fibrin deposits. Non-desmoplastic type includes replacement (r) and the pushing (p) pattern. In rHGP, the tumor permeates between the liver hepatocytes, without disruption of the normal architecture, while the tumor expands and compresses the surrounding hepatocytes in pHGP. Interestingly, the distribution of immune cells was different among three HGPs [12]. Increased cytotoxic CD8^+^ T cells, CD45^+^, CD79A^+^, Kappa/Lambda, and Self-ligand receptor of the signaling lymphocytic activation molecule family 7 (SLAMF7)^+^ cells and a higher CD8^+^/CD4^+^ ratio were observed in dHGP compared to other HGPs [13,14,15]. Additionally, multiple studies showed that desmoplastic type have improved prognosis compared to non-desmoplastic type [11,13,16,17,18].

Although the metastatic tumor of a small proportion of colorectal cancer liver metastasis patients can be removed by surgical resection, the 5-year survival rate is still, disappointingly, only around 36% [19], with 75% of patients undergoing rapid relapse after resection [20]. The survival rate after resection depends on a number of variables: liver metastasis, tumor size, node-positive primary cancer, preoperative carcinoembryonic antigen level. Some other additional therapies, like anti-epidermal growth factor receptor (EGFR) or anti-vascular endothelial growth factor (VEGF) (e.g., cetuximab, bevacizumab) have been used in treating CRC, but the financial burden is very high while the efficacy is still modest [21,22]. Immunotherapy has revolutionized the treatment of cancer, yet it is currently not widely applicable to CRC liver metastasis but show great potential in preclinical studies and clinical translation [23,24].

### 1.3. Pathogenesis of CRC Liver Metastases

Although cancer metastasis is the major cause of cancer related mortality, metastasis remains one of the poorly characterized aspects of cancer biology. Cancer metastasis is a sequential series of events called “metastatic cascade”, during which locally proliferating cancer cells acquire the invasive capability and translocate to the circulation, migrate to the distant organ, colonize, and form metastases eventually [25]. In CRC liver metastasis, mutations including KRAS, TP53, APC, PIK3CA, NRAS, BRAF, and SMAD4, etc., accompanied by genomic and epigenomic instability initiate CRC development and acquire the invasive phenotype [26]. Cooperating with immune cells via secretion of cytokines, chemokines, growth factors, and proteases, CRC cells reshapes the favorable tumor microenvironment to facilitate CRC liver metastasis [27]. The immune cells actively interact with CRC cells at every step of the metastatic cascade, including modulating tumor-infiltrating leukocytes to evade immune surveillance, formation of pre-metastatic niche, enhancing CRC extravasation and intravasation, protection of circulating or arrested CRC cells, promoting colonization, and reactivating dormant metastatic CRC cells in liver, which supports successful liver metastasis [28,29].

In this review, we summarize the underlying mechanisms of CRC liver metastasis facilitated by liver immune microenvironment in the process of liver metastatic cascade, especially the pre-metastatic niche formation and CRC colonization and propagation in liver. Furthermore, we discuss how the reciprocal interaction between immune cells and CRC cells influence liver metastasis formation and response to therapy, and discuss the potential of therapies that target the liver immune microenvironment to treat CRC liver metastasis.

## 2. Liver Immune Microenvironment for CRC Liver Metastasis

The unique and complex microenvironment of liver with enriched vessel permeability and dampened immune response to neoantigens makes it a fertile soil for cancer cell metastasis. In this section, we will review the hepatocyte-derived factors, non-parenchymal cells, and various immune cells that cooperate in the liver to form a metastasis-prone microenvironment for CRC cells.

### 2.1. Hepatocyte-Derived Factors

#### 2.1.1. Inflammatory Cytokines/Chemokines and Growth Factors

A substantial number of studies has demonstrated that tumor derived factors including cytokines and chemokines drive pre-metastatic niche formation in the distant organ to support the incoming of metastatic tumor cells. The premetastatic niche protects tumor cells from immune attack by cytotoxic lymphocytes, which nullifies the efficacy of immunotherapy and facilitates metastasis [30,31,32]. In fact, not only tumor cells but also parenchymal hepatocytes play a role in regulating liver metastasis. Recently, an interesting study showed that during early pancreatic cancer development, non-malignant cells secreted IL-6 to activate signal transducer and activator of transcription 3 (STAT3) signaling in hepatocytes. Subsequently, these hepatocytes produced serum amyloid A1 and A2 (SAA) to induce myeloid cell accumulation and alter the fibrotic microenvironment in the liver to establish the pre-metastatic niche. Consistently, overexpression of SAA and activation of STAT3 were observed in the liver of pancreatic cancer and colorectal cancer liver metastasis patients. In addition, circulating SAA levels were significantly higher in liver metastasis patients, which correlated with poor survival [33]. Another study also showed that the expression of hepatic cytokines (tumor necrosis factor α (TNF-α), IL-1 beta, IL-6, IL-10) and other factors noted to be involved in the colonization of CRC cells including intercellular adhesion molecule 1 (ICAM-1), chemokine (C-C motif) ligand 2 (CCL-2), CCL-7, matrix metalloproteinase-2 (MMP-2), and MMP-9 were significantly increased in alcohol-injured liver, and positively correlated with rate and burden of CRC liver metastases [34]. Moreover, hepatocytes release multiple factors, such as insulin-like growth factor 1 (IGF-1), hepatocyte growth factor-like protein/macrophage stimulating-protein (HGFL), and hepatocyte-derived heregulin (HRG), which can induce tumor cell growth, invasion, and metastasis through different mechanisms [35,36,37]. It was also reported that tumor activated hepatocyte and myofibroblast could affect the phenotype of primary CRC cell by upregulating liver metastatic gene expression (e.g., *S100P*, *cadherin-H1*, *osteopontin, transforming growth factor beta (TGF-β)*, *thioredoxin-1*). For example, TGF-β induced the expression of extracellular matrix protein by colon cancer cells, which enhances their aggressiveness and metastatic properties [38] (Figure 1).

#### 2.1.2. Cyclin-Dependent Kinases (CDKs)

CDKs play pivotal roles in the regulation of cell division and transcription in response to extra- and intracellular cues and deregulation of CDKs is a hallmark of cancer [39]. A recent case report showed that a CDK4/6 inhibitor together with hormonal therapy successfully managed visceral metastases and provided long-term survival in a patient with breast cancer liver metastases [40]. CDK8 has been reported to be overexpressed in colon cancer, and inhibition of CDK8 did not affect colon cancer cell growth but significantly suppressed colon cancer liver metastasis. Mechanistically, CDK8 downregulated the expression of TIMP metallopeptidase inhibitor 3 (TIMP3) via TGFβ/SMAD-driven expression of a TIMP3-targeting microRNA, miR-181b, which consequently increased MMP expression. In addition, CDK8 induced Mmp3 transcription in murine or MMP9 in human colon cancer cells through Wnt/β-catenin signaling pathway [41]. MMPs play pivotal roles in various biological processes, including matrix degradation, angiogenesis, cell adhesion, growth factor receptor signaling, apoptosis, ECM remodeling, and immune regulation, which facilitated cancer progression and metastasis [42].

Our previous study also reported that cell cycle-related kinase (CCRK, also called CDK20) activated nuclear factor-kappa B (NF-κB) signaling in hepatocytes (parenchymal cells) to increase the polymorphonuclear myeloid-derived suppressor cell (PMN-MDSC)-trafficking chemokine C-X-C motif ligand 1 (CXCL1) expression in the liver. Increased CXCL1 recruited PMN-MDSC and reduced natural killer T (NKT) cells in liver to form the pre-metastatic niche for melanoma and colorectal cancer liver metastasis in CCRK transgenic mice. Accordingly, CRC liver metastasis patients exhibited hyperactivation of hepatic CCRK/NF-κB/CXCL1 signaling, which was associated with accumulation of PMN-MDSCs and a paucity of NKT cells compared to healthy liver transplantation donors [43] (Figure 1). This study highlighted the parenchymal-immune cell crosstalk in shaping the liver immune microenvironment for CRC liver metastasis.

Collectively, these studies demonstrated that in addition to tumor derived factors, hepatocytes also participated in the formation of pre-metastatic niche for cancer metastasis.

### 2.2. Liver Non-Parenchymal Cells

In addition to hepatocyte-derived factors, increasing evidence suggests that non-parenchymal cells, such as liver resident fibroblast hepatic stellate cells (HSCs), liver resident macrophage Kupffer cells (KCs), liver sinusoidal endothelial cells (LSECs), and liver infiltrating immune cells around the hepatocytes also have critical roles in multiple stages during the development of CRC liver metastases either by direct or indirect cell-to-cell interaction.

#### 2.2.1. Hepatic Stellate Cell (HSC) and Cancer Associated Fibroblasts (CAF)

HSCs, also known as perisinusoidal cells, are resident pericytes localized in the perisinusoidal space of Disse, accounting for ∼10% of all resident cells in liver [44]. HSCs have various functions in normal and injured liver, and play a pivotal role in premetastatic niche. Our previous study showed that activated HSCs induced monocyte-intrinsic p38 MAPK pathway to trigger enhancer reprogramming for M-MDSC differentiation and immunosuppression, indicating the non-parenchymal-immune cell crosstalk in HCC development. In addition, the accumulation of M-MDSC in fibrotic liver was associated with reduced cytotoxic T cells and HCC progression [45]. Besides, a study also showed that pancreatic ductal adenocarcinoma (PDAC)-derived exosomes containing macrophage migration inhibitory factor (MIF) were taken up by KCs and subsequently activated resident HSC via TGF-β, leading to upregulation of fibronectin (Figure 2, in space of disse). This fibrotic environment further recruited tumor associated macrophages in liver premetastatic niche and facilitated the adhesion of disseminated tumor cells [31]. In addition, activated HSC secreted periostin to enhance CRC and endothelial cell survival in liver via the αvβ3 Integrin-Akt/PKB pathway [46]. Co-injection of HSC with CRC cells significantly promoted liver metastasis by enhancing angiogenesis [47].

Meanwhile, as a type of fibroblast, activated HSCs also preserve the metastatic-prone features of common cancer associated fibroblasts (CAF). CAF is one of the most abundant stromal cells in the tumor microenvironment, which can promote tumor growth, angiogenesis, and metastasis. In CRC, the distribution of CAFs in primary CRC has been demonstrated to be associated with malignant potential and prognosis of CRC patients [48]. Fibroblast activation protein-α (FAP) derived from CAFs was reported to be related to liver metastasis and poor clinical outcome [49]. Mechanistically, CAFs facilitated liver metastasis by supporting CRC cells’ adhesion and promoting CRC cell stemness and drug resistance. CAFs secreted hepatocyte growth factor (HGF) to induce CD44 expression on CRC cells via HGF/MET/AKT signaling, which promoted adhesion and migration of CRC cells [50]. Moreover, TGFβ1 induced adhesion of CRC cells to CAFs, and co-migration of CAFs and CRC cells remarkably enhanced liver metastasis [51]. CAFs can also directly transferred exosomes (containing miR-92a-3p) to CRC cells, which subsequently activated Wnt/β-catenin pathway and inhibited mitochondrial apoptosis, contributing to cell stemness, EMT, metastasis, and 5-FU/L-OHP resistance in CRC [52]. Reciprocally, CRC cells also induced and modified CAFs to facilitate metastasis. CRC cells activated HSCs and induced their differentiation into CAFs [53]. Similarly, elevated carcinoembryonic antigen (CEA) level by CRC cells activated and transformed fibroblast to CAF phenotype, which remodeled the extracellular matrix and promoted CRC cells adhesion and liver metastasis [54]. In addition, it was reported that the crosstalk between SMAD4 deficient CRC cells (instead of SMAD4 proficient CRC cells) and CAFs induced bone morphogenetic protein 2 (BMP2) expression in CAFs and consequently promoted CRC invasiveness and liver metastasis in preclinical model [55]. Dysregulation of BMP signaling in CAF predicted and modified CRC progression and prognosis. Targeting CAF by regulation of BMP signaling reduced CRC liver metastasis [56].

#### 2.2.2. Liver Sinusoidal Endothelial Cell (LSEC)

LSECs are the major resident non-parenchymal cells in liver, which line the low shear, sinusoidal capillary channels of the liver. LSECs have vital immunological functions like antigen presentation, leukocyte recruitment, and physiological functions like filtration and endocytosis [57]. When the disseminating tumor cells arrive at the liver through the circulation, they are arrested and trapped in the sinusoidal capillaries in liver. Here, some tumor cells are killed by the immunosurveillance of tissue resident KCs and NK cells [58] and the remaining surviving tumor cells extravasate into the perisinusoidal space (space of Disse). In this process, LSECs upregulate the expression of cell adhesion molecules, such as vascular cell adhesion molecule 1 (VCAM-1), intercellular cell adhesion molecule 1 (ICAM-1), and E-selectin to support the arrest, retention, and transmigration of the tumor cells. Multiple preclinical studies showed that inhibition of adhesion molecules by targeting integrin β2 (ligand of ICAM-1) [59], blockade of adhesion molecules [60], and disruption of inflammatory TNFα/TNF receptor 2 (TNFR2) signaling [61,62], Notch signaling [63] reduced liver metastasis.

In addition, the reciprocal crosstalk between tumor cells and LSECs also enhance survival and metastatic potential of tumor cells and promote angiogenesis. Ligands such as CD44, sLewA, and sLewX expressed on tumor cells interacted with E-selectin on inflamed LSECs, which promoted CRC liver metastasis [64,65] and further increased adhesion molecule expression on LSECs by upregulating high-mobility group box 1 (HMBG1) release [66]. Simultaneously, LSECs secreted fibronectin and macrophage migration inhibitory factor (MIF) that could induce EMT phenotype in CRC cells resulting in increased invasion and migration of CRC cells into the liver parenchyma [67] (Figure 2, in sinusoid/space of disse).

#### 2.2.3. Kupffer Cell

Kupffer cells (KC) are resident macrophages in the liver, which play a dual role in the tumor microenvironment of liver metastasis. On one hand, KCs exert tumoricidal activity by phagocytosis, releasing oxygen metabolites, cytotoxic cytokines, and secreting proteases [68,69,70,71]. During the initial stage of CRC liver metastasis, KCs secrete TNF-α in liver, contributing to metastasis control [61]. On the other hand, KCs can induce cell adhesion molecule expression on LSECs, which helps the adhesion of disseminated tumor cell arrest in liver, and produces factors (e.g., IL-6, MMPs, VEGF, etc.) that promote tumor cell invasion, proliferation, and angiogenesis. In the tolerant state, KCs can also release inhibitory cytokine IL-10, induce regulatory T cells (Tregs), and express T cell inhibitory molecule programmed cell death 1 ligand (PD-L1), which ameliorates anti-tumor immunity [72] (Figure 2, in sinusoid/space of disse). Collectively, these studies highlighted a complex non-parenchymal/immune cell-immune cell crosstalk in liver microenvironment.

### 2.3. Liver-Infiltrating Immune Cells

#### 2.3.1. Neutrophil and Myeloid-Derived Suppressor Cell (MDSC)

Neutrophils are innate immune cells and show functional plasticity driven by multiple factors in cancer, depending on different microenvironments [73,74]. Various factors have been demonstrated to support the recruitment and accumulation of neutrophils and PMN-MDSC in liver metastases. It was shown that tumor derived tissue inhibitor of metalloproteinases (TIMP)-1 level was increased in CRC patients and correlated with liver metastasis [75]. Mechanistically, TIMP1 upregulated stromal-derived factor (SDF) 1 to recruit neutrophils to the liver, which facilitated CRC liver metastasis. Inhibition of SDF-1/CXCR4 axis or depletion of neutrophils significantly reduced liver metastasis in mice [75]. Additionally, lysyl oxidase-like 4 (LOXL4) protein was demonstrated to be upregulated in neutrophils in CRC liver metastases with replacement HGP compared to desmoplastic type of liver metastases and the adjacent normal liver, which was associated with resistance to neoadjuvant anti-angiogenic therapy [76]. The expression of LOXL4 was significantly higher in circulating neutrophils of these patients compared with healthy control, and can be induced by stimulation with lipopolysaccharide and TNF-α. These studies suggested the multifunctional role of neutrophils in liver metastases. Another study showed that CRC cells overexpressed VEGF-A to induce CXCL1 secretion from macrophages, which subsequently recruited CXCR2 positive MDSC in liver to form the metastatic niche [77]. Similarly, it was showed that secretion of OPN, MMP9, S100A8, S100A9, SAA3, and VEGFA were increased in a CT26FL3 liver metastasis mouse model to enhance bone marrow derived-cell recruitment in liver for pre-metastatic niche formation [78]. Our previous study also found that CCRK-CXCL1 mediated PMN-MDSC recruitment in liver and reduced NKT cell infiltration were significantly correlated with melanoma and CRC liver metastasis. Inhibition of PMN-MDSC restored NKT cell infiltration and ameliorated liver metastasis [43]. In a mouse colon cancer and lung cancer liver metastasis model, accumulation of MDSC was associated with liver metastasis dependent on TNFR2 signaling. Disruption of TNFR2 signaling significantly reduced MDSC accumulation and liver metastasis [62]. Clinically, it was also demonstrated that circulating MDSC level was positively correlated with metastatic tumor burden in various types of solid tumors [79].

In addition, neutrophils can also promote liver metastasis in an immune-independent manner. Neutrophils arrested on LSEC in liver sinusoids increased tumor cell adhesion by interaction of CD11b/CD18 (Mac-1) on neutrophils and ICAM on tumor cells, acting as a bridge between disseminating tumor cells and liver parenchyma [80] (Figure 2, in sinusoid/space of disse/parenchyma).

#### 2.3.2. Monocyte, Tumor Associated Macrophage (TAM) or Metastasis-Associated Macrophage (MAM)

Macrophages are plastic and can polarize to tumoricidal or pro-tumorigenic macrophages under different microenvironments. In liver metastases, monocytes and TAMs or MAMs were recruited in liver by tumor cells via CCL2 secretion. And adoptive transfer of inflammatory monocytes preferentially migrated to the metastatic sites and differentiated into MAMs [81]. Recruitment of macrophages in the liver facilitated liver metastasis by inducing liver fibrosis and immunosuppression [82,83]. CCR2 antagonists or knockout of CCL2 in tumor cells significantly reduced metastatic tumor burden [81,84,85]. In a mouse CRC liver metastasis model, loss of Ndrg2 (N-myc downstream-regulated gene 2) gene in macrophage shifted TAM polarization to M1 phenotype and thus alleviated CRC liver metastasis [86] (Figure 2, in sinusoid/space of disse/parenchyma). However, it is difficult to differentiate the effect of inhibiting tumor cell seeding or growth in most studies using animal models, when reduced liver metastasis was observed.

#### 2.3.3. NK Cell

NK cell accounts for 50% of the liver lymphocyte population and exhibits anti-tumor function mediated by the release of cytotoxic granules, TNF-related apoptosis-inducing ligand (TRAIL) and Fas ligand (FasL) [87]. As part of innate immunity, NK cells can exert killer function towards transformed and stressed cells. Besides, NK cells can also modulate innate and adaptive immunity by secretion of chemokine and cytokine [88,89]. Substantial evidence shows that NK cell plays a pivotal role in controlling cancer metastasis [90]. In a preclinical mouse CRC liver metastasis model, NK cells were demonstrated to restrain CRC liver metastasis. However, the function of NK cells was impaired in hepatic metastases compared to NK cells in healthy livers. More interestingly, the differentiation of NK cells was instructed by signals from the liver microenvironment bearing metastatic tumors, indicating the complex crosstalk between NK cells and CRC liver metastases [91]. Further mechanistic study revealed that CRC liver metastases produced lactate to modulate the pH of the tumor microenvironment, which induced mitochondrial stress and apoptosis of liver-resident NK cells migrating towards the tumor, leading to metastases outgrowth [92]. In addition, nucleotide-binding oligomerization domain family pyrin domain containing 3 (Nlrp3) inflammasome-IL-18 pathway could also regulate the maturation, surface expression of the death ligand FasL, and tumoricidal activity of hepatic NK cells. Thus, Nlrp3 deficiency significantly impaired effective NK-cell-mediated tumor attack required to suppress CRC liver metastasis [93]. Moreover, TRAIL-expressing NK cells were proved to be important in suppressing liver metastasis. Neutralization of TRAIL using monoclonal antibody abolished NK cell-mediated metastasis control [94]. Clinically, it was shown that NK cell infiltration combined with CD8^+^ T cells has enhanced the prognosis of CRC patients, indicating a potential supporting role for NK cells in the anti-CRC effects of CD8+ T cells [95].

#### 2.3.4. NKT Cell

NKT cells are innate-like lymphocytes that share properties of both NK cells and T cells. Similar to NK cells, NKT cells can exert both anti-tumor killer function and modulate immune responses by secretion of cytokines [96]. It has been reported that CXCL16 could promote NKT cell liver infiltration to potently suppress CRC liver metastasis in vivo [97]. Another study further pointed out that gut microbiome used bile acids as a signal to regulate LSEC-derived CXCL16 which recruited NKT cells in liver [98]. Consistently, our recent study demonstrated that CCRK-CXCL1-MDSC axis activation promoted CRC liver metastasis through suppression of anti-tumor hepatic NKT cells, while depletion of MDSC could restore hepatic NKT cells and reduce CRC liver metastasis [43]. In comparison, some controversial studies also pointed out that NKT cells exacerbated liver metastasis arising from intraocular melanomas by inhibiting the anti-tumor activity of liver NK cells [99]. Therefore, further studies were needed to investigate the different subsets and functions of NKT cells in the context of liver metastasis derived from different cancer types.

#### 2.3.5. Regulatory T Cells (Tregs) and Other Cells

CD4^+^FoxP3^+^ Tregs are immunosuppressive cells that suppressed effector T cell functions. It has been reported that increased accumulation of Tregs dependent on TNFR2 signaling correlated with colon and lung cancer liver metastasis. Genetically deficient for TNFR2 or TNFR2 antisense oligodeoxynucleotides significantly reduced Tregs and MDSC accumulation and decreased liver metastasis [62]. In a retrospective study, high Treg infiltration predicted poor clinical outcome of CRC liver metastasis patients, suggesting that infiltrating Treg cells support the growth of established CRC liver metastases [100].

In addition, bone marrow-derived VEGFR1-positive progenitors were also recruited to the pre-metastatic niche, and then formed clusters and promoted the adherence and growth of subsequently disseminating tumor cells [101]. These immature myeloid cells also secreted MMP9 to promote tumor cell invasion to the parenchyma [102].

### 2.4. Role of Extracellular Matrix (ECM)

ECM is a structural scaffold comprised by dynamic macromolecules and their regulatory factors [103], which can support outgrowth and treatment resistance of the arrived tumor cells. Proteomic analysis of three sequential CRC liver metastases in one patient found different ECM phenotypes for recurrent metachronous metastases, associated with different grades of malignancy [104]. Different components of ECM have been studied. In CRC liver metastasis patients, fibroblasts in liver increased ECM stiffening, which enhanced angiogenesis and promoted drug resistance of anti-angiogenic therapy. Reduction of stiffness largely increased the efficacy of anti-angiogenic therapy [105]. Similarly, preoperative treatment with anti-VEGF therapy markedly enhanced hyaluronic acid (HA, component of ECM) deposition within the tumors. Preclinical models demonstrated that hypoxia drove the remodeling of the ECM and thus increased tumor stiffness and reduced drug perfusion in liver metastases. Depletion of HA could reduce the physical barriers to systemic treatments in CRC liver metastases [106].

Neutrophil extracellular traps (NET), also a component of ECM, are comprised of extracellular fibres, primarily webs of DNA, with associated proteolytic enzymes secreted by neutrophils in response to inflammatory cues that trap and kill invading pathogens. Emerging evidence showed that NET can sequester tumor cells arriving at liver, increase their retention, promote tumor cell proliferation and migration, and thus facilitate metastasis [107,108]. It was shown that NET-like structures around metastatic breast cancer cells were observed in both the lungs of mice and clinical breast cancer specimens. Inhibition of NET formation or digestion of NET markedly reduced metastasis [109].

### 2.5. Immune Checkpoint Molecules

Immune checkpoint molecules such as programmed cell death protein 1 (PD-1), cytotoxic T lymphocyte-associated antigen-4 (CTLA4), and T cell immunoglobulin and mucin-domain containing-3 (Tim-3), are negative regulators of the immune system to prevent self-attack. However, this mechanism is utilized by cancers to escape from anti-tumor immunity. Previous study showed that the expression of PD-L1 was increased in liver metastases compared to primary CRC, indicating different intrinsic microenvironment between primary and metastatic CRC [110], which may help CRC liver metastases escape from immune surveillance. In addition, it was reported that chemotherapy can modulate PD-L1 and TIM-3 expression in CRC liver metastases, suggesting the potential strategy of combined chemo-immunotherapies [111]. Preclinical study showed that dual CTLA4 and PD-1 blockade could significantly suppress colon cancer growth and liver metastasis by enhancing T cell responses and M1 macrophage polarization [112].

## 3. Therapeutics for CRC Liver Metastasis

### 3.1. Current Therapies for CRC Liver Metastasis

Over the past few decades, advancements have been made in understanding the potential mechanisms and developing therapies for cancer liver metastasis. Although it has been greatly improved, the overall survival of cancer liver metastasis patients remains low. It is difficult to cure the cancers once they metastasize to other organs. The present therapeutic strategies in use for eradicating metastatic tumors are fundamentally the same as treatment targeting primary tumors. For CRC liver metastasis patients, current therapies for liver metastasis are surgical resection, systemic and localized therapies, and combination regimen is also frequently used.

#### 3.1.1. Surgical Resection

Based on multiple retrospective and comparative studies, surgical resection remains the gold standard in treating CRC liver metastasis patients and provides long-term survival [19,113,114,115]. There are two strategies of surgical resection, namely, simultaneous resection and staged resection. But no significant statistical difference on survival was observed between two types of resection [116]. Generally, patients with good liver function and general condition and without metastasis in other organs except liver, are suitable for surgical resection. In particular cases, when CRC patients developed both liver metastases and small lung metastases, liver metastases can still be resected with lung metastases resected or ablated synchronously/metachronously. In a retrospective study, 99 (16%) of 612 patients survived for ten years after hepatic resection, while 34% of the 5-year survivors succumb to cancer related death [117]. Although surgical resection provides better survival, only a minority of patients are resectable when diagnosed [118]. In addition, more than 50% of patients will still develop local and distant recurrence after resection [117,119].

#### 3.1.2. Systemic Therapy

For patients with extensive colorectal cancer metastases to both liver and other organs, systemic chemotherapy is a more appropriate option. The combination of oxaliplatin or irinotecan plus leucovorin and 5-fluorouracil is a frequently used chemotherapy regimen that could significantly improve disease outcome in CRC liver metastasis patients [120,121,122]. Drugs that target epithelial and vascular endothelial growth factor pathways, such as cetuximab and bevacizumab, are also used to treat these patients [21,22].

Moreover, emerging strategies have been designed to increase the resectability of CRC liver metastasis patients. In these patients with initially unresectable liver metastases, treatment of neoadjuvant chemotherapy may not cure the disease but downstage the tumor, which provides an opportunity for resection. Numerous studies in this field endeavor to increase the eligibility for resection, refining the indications and contraindications for surgery, and improving patient survival [123]. The National Comprehensive Cancer Network (NCCN) guidelines recommend FOLFOX (folinic acid plus fluorouracil and oxaliplatin), FOLFIRI (folinic acid and short-term infusional fluorouracil plus irinotecan), or XELOX (capecitabine and oxaliplatin; also called CAPOX) with or without bevacizumab; FOLFIRI with or without cetuximab or panitumumab; or FOLFOX with or without panitumumab or cetuximab (if RAS wild type) (https://www.nccn.org/professionals/physician_gls/ (accessed on 24 March 2021).). Data showed that 12.5% of unresectable CRC liver metastasis patients acquired the opportunity to liver resection by chemotherapy and had longer survival with lower operative risk [124,125].

#### 3.1.3. Localized Therapy

Localized therapy including radiofrequency ablation (RFA), hepatic artery catheter chemotherapy and chemoembolization and portal vein embolization (PVE), radiation therapy, are used in patients with unresectable liver metastases without extrahepatic diseases [126]. Among all the treatment modalities, RFA is more frequently used, for its minimal invasiveness with lower mortality rate, fewer complications, reduced hospital days, and costs compared to other therapies. RFA may not cure the disease for most CRC liver metastasis patients, but relieve or control the disease and improve the quality of life of unresectable patients [126]. Hepatic artery catheter chemotherapy and chemoembolization and portal vein embolization can also be considered as alternative treatments to systemic chemotherapy, which can increase drug delivery in the liver but reduce systemic toxicity and showed improved response rate compared to conventional systemic chemotherapy [127]. Selective internal radiation therapy (SIRT) delivering 90Yttrium microspheres to the hepatic artery is another alternative choice for CRC liver metastasis patients. This treatment achieved a high response rate and encouraging survival in CRC liver metastasis patients [128].

### 3.2. Therapeutic Perspectives

#### 3.2.1. Targeted Therapy Development, e.g., CDKs

Although liver metastasis accounts for most cancer related mortality in CRC patients, the underlying mechanisms driving this disease progression remain largely unknown, leading to the lack of effective therapy. A recent case report showed that CDK4/6 inhibitors exhibited encouraging clinical outcome in treating metastatic breast cancer and colon cancer [40]. Concordantly, our recent studies demonstrated that as the latest family member of CDK, CCRK is the novel signaling hub exploitable in liver disease [129]. Upregulation of CCRK was observed in multiple cancers, such as HCC and colon cancer, which correlated with tumor staging and poor survival and prognosis [130]. In hepatocellular carcinoma (HCC), CCRK mediates tumor development in different etiologies, including hepatitis B virus infection [131], non-alcoholic fatty liver disease [132], via orchestrating a self-reinforcing circuitry comprising of AR, GSK3β, β-catenin, AKT, EZH2, and NF-κB signaling, and also facilitate tumor immune evasion [133,134,135]. Knockout *C**crk* in mouse hepatoma significantly enhances the efficacy of immune checkpoint inhibitor by disrupting immunosuppression and unleashing anti-tumor immune response [135,136]. In addition, the ectopic expression of CCRK induced by chronic inflammation in liver shapes the immune microenvironment by accumulating immunosuppressive PMN-MDSCs and reduced anti-tumor NKT cells to facilitate CRC liver metastasis [43]. Collectively, CCRK plays a pivotal role in regulating liver microenvironment. Thus, modulation of CCRK may be potential therapy for CRC liver metastasis. Given the ability to design inhibitors of a number of the CDK enzyme family (most notably CDK4/6 as highlighted above), then development of a selective inhibitor to target CCRK would be both a feasible and attractive drug development opportunity.

#### 3.2.2. Immunotherapy Development

Multiple treatment modalities have been developed to treat CRC liver metastasis; however, the efficacy remains unsatisfying. Most patients relapse after these treatments and succumb to the disease; therefore, more effective therapeutic strategies are desperately needed. In addition, development of effective adjuvant therapy after curative treatment of primary CRC tumor to prevent liver metastasis may be of great clinical significance based on the high occurrence of CRC liver metastasis.

Immunotherapy, including immune checkpoint inhibitors (ICIs), cancer vaccines, and chimeric antigen receptor (CAR) T cell therapy, has achieved promising results in many cancers and revolutionized the treatment of cancer by enhancing anti-tumor immune responses. A recent study showed that anti-PD-L1 monotherapy in patients with metastatic or unresectable CRC with mismatch repair deficiency (dMMR)/microsatellite instability-high (MSI-H) displayed remarkable anti-tumor activity with manageable toxicity [23]. Additionally, a case report showed that patients who progress on anti-programmed cell death protein 1 (PD-1) therapy still respond to combinatory immunotherapies (nivolumab plus ipilimumab) [137]. Another phase I clinical trial escalating dosage of CAR-T therapy in metastatic CRC patients also observed potential treatment response in some patients without severe adverse events [24]; however, the efficacy remains limited owing to the inhibitory impact of the tumor immune microenvironment, and many patients still cannot benefit from these treatment [138,139].

As discussed above, the immunosuppressive microenvironment in liver contributes to the generation of pre- and pro-metastatic niches to facilitate cancer liver metastasis development, immune evasion, and affect treatment response. A preclinical study showed that CAR T-cell therapy in conjunction with reduction of Tregs and MDSCs hindered CRC growth [140,141], indicating that combination of immunotherapy and targeting the immunosuppressive immune microenvironment may be potentially effective in treating CRC liver metastasis or as adjuvant therapy to prevent liver metastasis.

### 3.3. Current Clinical Trials

Several therapeutic strategies are under investigation in clinical trials, including chemotherapy, targeted therapy, radiotherapy, ablation, surgery, immunotherapy, and more frequently, the combination of a few therapies (Figure 3). Current ongoing clinical trials for CRC liver metastasis are listed in Table 1. Chemotherapy (either systematically or locally administered) combined with other therapies remains the major strategy in clinical trials. Many studies reported that neoadjuvant chemotherapy could convert unresectable liver metastases to resectable tumors, which improved clinical outcomes [142,143]. Interestingly, immunotherapy such as CAR-T cell or modified T cell therapy, immune checkpoint inhibitor (ICI), TLR agonist, and Granulocyte-macrophage colony-stimulating factor (GM-CSF) are also being investigated in several clinical trials (Table 1). In addition, other therapies, such as ultrasound mediated local therapy, Vitamin D3, and ATP128 (a self-adjuvanted chimeric recombinant protein vaccine) are under investigation.

## 4. Conclusions

In this review, we emphasized on the critical role of the premetastatic niche in the liver microenvironment in facilitating cancer liver metastasis. The disseminating tumor cells depend on interaction with the liver immune microenvironment for arrest, immune evasion, colonization, migration, and proliferation. Thus, a better understanding of the molecular mechanisms orchestrating the formation of a hospitable hepatic metastatic niche and the identification of the drivers supporting this process is critical for the development of better therapies to stop or at least decrease liver metastasis. Besides, the anatomic proximity between liver and colon as well as specific signals derived from CRC cells may partially explain the clinical preference of CRC liver metastasis. Nevertheless, CRC liver metastasis may also share similar mechanisms with the liver tropism of different cancer metastasis via the regulations of liver immune microenvironment, as we summarized here. Therefore, mechanistic insights on CRC liver metastasis may also pave ways to new perspectives in liver metastasis from other cancer types.

Traditional chemotherapy, surgical resection, and localized therapy still dominate the treatment for CRC liver metastasis. Although these therapies can remove or control the metastatic tumors, recurrence remains the major challenge for successful management of the disease. Immunotherapy is characterized by strong and long-lasting effects with fewer side effects. Based on the importance of hepatic niche in every step of liver metastasis, targeting the immune microenvironment by immunotherapy will be potential in treating CRC liver metastasis. Future studies should use preclinical models or single cell sequencing to investigate the complex immune crosstalk in the liver microenvironment to identify the potential target and translate to the clinics.

## Figures and Tables

**Figure 1 cancers-13-02418-f001:**
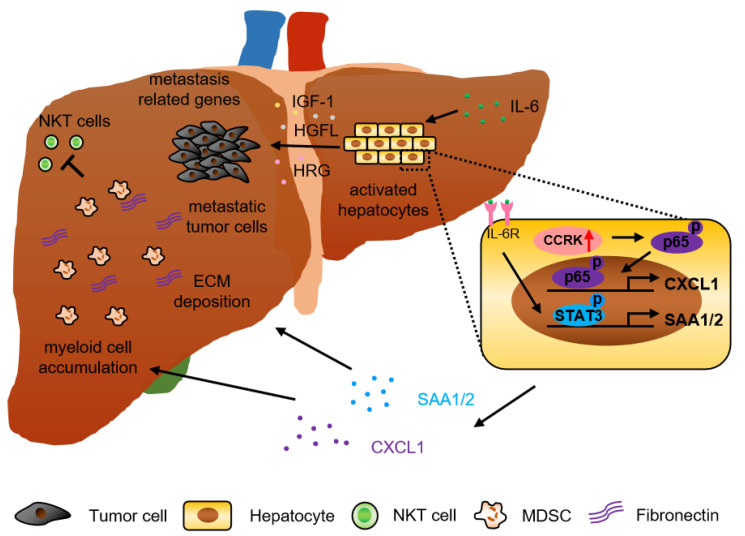
Hepatocyte-derived factors drive pre-metastatic niche formation for cancer metastasis. Schematic diagram showing the effects of hepatocyte-derived factors on establishment of pre-metastatic niche in CRC liver metastasis. Hepatocytes activated by primary tumor cells upregulate CXCL1 and SAA1/2 expression to increase myeloid cell accumulation and ECM deposition, which facilitates cancer liver metastasis. In addition, these hepatocytes also secrete multiple factors like IGF-1, HGFL, HRG to promote metastatic tumor growth.

**Figure 2 cancers-13-02418-f002:**
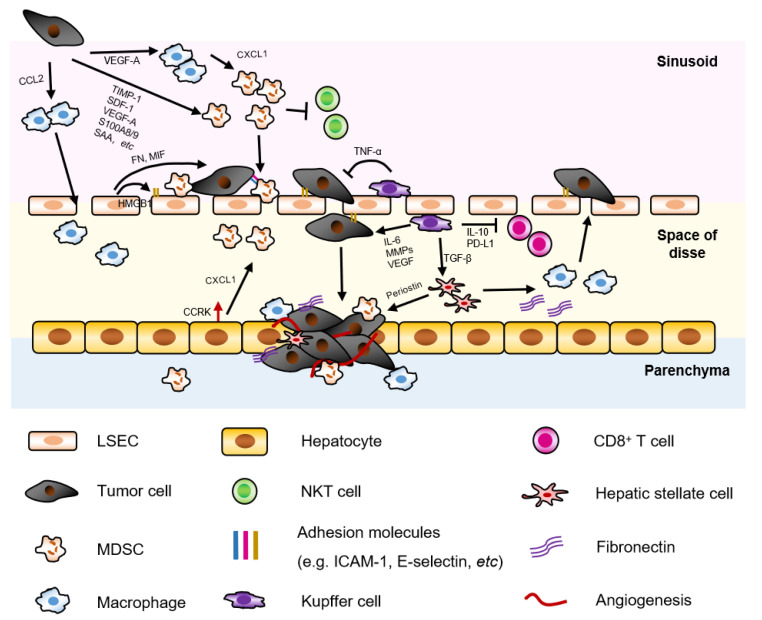
Immunoregulation of disseminated tumor cells by metastatic-prone liver immune microenvironment. The successful colonization of colorectal cancer cells in liver depends on the interaction between tumor cells and the liver immune microenvironment. The primary tumor cells secrete multiple factors to recruit MDSCs and macrophages in liver, which suppresses NKT cells, facilitates the arrest and invasion of the tumor cells and promotes angiogenesis. LSECs support the arrest, retention, and transmigration of tumor cells by the expression of adhesion molecules, and secrete fibronectin (FN) and MIF to induce EMT phenotype in CRC cells. Kupffer cells can both inhibit tumor cell growth by secretion of TNF-α and support tumor cell metastasis via suppression of CD8+T cells, activation of HSC, and promoting tumor cell invasion. The activated HSCs upregulate fibronectin, recruit macrophage to promote adhesion of disseminating tumor cells, and secrete periostin to enhance angiogenesis.

**Figure 3 cancers-13-02418-f003:**
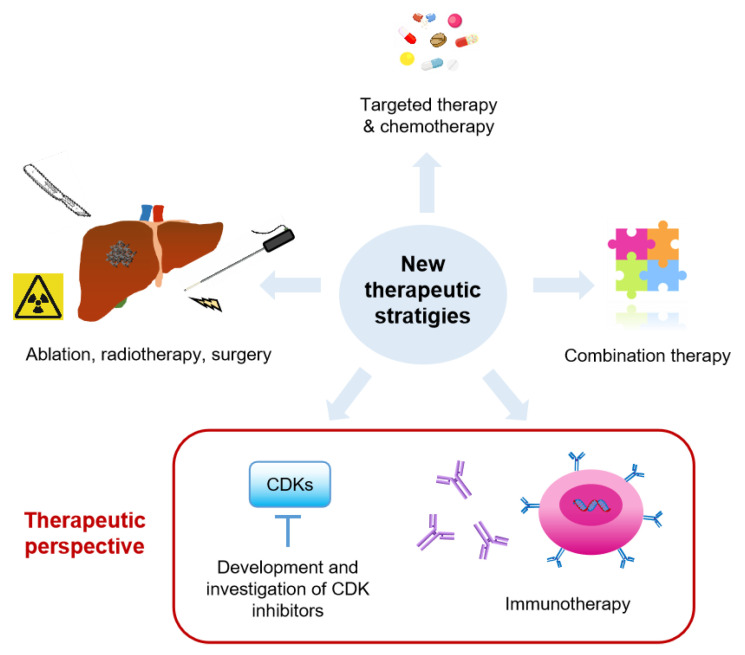
Therapeutic perspectives for colorectal cancer liver metastasis. Several new therapeutic strategies are under clinical trials, including: (1) novel or the combination of targeted therapy and chemotherapy, either locally or systemically; (2) ablation, radiotherapy, surgery; (3) the combination of current therapies; (4) development and investigation of CDK inhibitors; (5) immunotherapy like immune checkpoint inhibitor, CAR-T cells, etc.

**Table 1 cancers-13-02418-t001:** Ongoing clinical trials evaluating the use of different modalities in metastatic colorectal cancer.

Trial ID	Phase	Study Population	Intervention	Recruitment Status
**Targeted therapy and chemotherapy (Locally or systemically)**				
NCT04509635	III	Colorectal Cancer Liver Metastasis with Progression After First-line Treatment of Cetuximab	Cetuximab±Chemotherapy	Not yet recruiting
NCT03493061	II	Unresectable Colorectal Cancer Liver Metastases	Irinotecan; Oxaliplatin; Floxuridine	Recruiting
NCT04189055	II	Neo Wild-type RAS/RAF Metastatic Colorectal Cancer with Liver Metastases	Cetuximab; Irinotecan	Recruiting
NCT04276090	NA	Unresectable Colorectal Metastases/Intrahepatic Cholangiocarcinoma	Codman Catheter; Synchromed Pump Hepatic Artery Chemotherapy	Recruiting
NCT03697044	NA	Colorectal Cancer Liver Metastases	Irinotecan Drug-Eluting-Bead Trans Arterial ChemoEmbolisation	Not yet recruiting
NCT03125161/ NCT02102789	III	Unresectable Colorectal Cancer Liver Metastases	HAI; Chemotherapy ± target therapy or mFOLFOX6	Recruiting
NCT04003792	II	Unresectable Colorectal Cancer Liver Metastases	oxaliplatin; FOLFIRI Protocol; Bevacizumab	Recruiting
NCT00695201	I	Unresectable Colorectal Cancer Liver Metastases	Floxuridine, Oxaliplatin, CPT-11	Active, not recruiting
NCT04194034	I/II	Unresectable Colorectal Cancer Liver Metastases	TG6002; Flucytosine (5-FC)	Recruiting
NCT04595266	II	Colorectal Cancer Liver Metastases	FOLFOX regimen; Anti-EGFR or Bevacizumab; LIVERPEARLS-Irinotecan	Not yet recruiting
NCT03164655	II	Unresectable Colorectal Cancer Liver Metastases	Oxaliplatin, Cetuximab, Bevacizumab, Panitumumab, Irinotecan, Leucovorin, 5-Fluorouracil	Recruiting
NCT03031444	II/III	Resectable Colorectal Liver Metastasis	Cetuximab plus FOLFIRI/FOLFOX; FOLFIRI/FOLFOX/CapeOX	Recruiting
NCT04525326	III	Unresectable Colorectal Cancer Liver Metastases	Cetuximab; Bevacizumab; mFOLFOX/FOLFIRI (Standard Chemotherapy)	Not yet recruiting
NCT03366155	II	Colorectal Cancer Liver Metastases	Panitumumab; FUDR-Dex; Oxaliplatin; 5FU; Irinotecan; cetuximab	
NCT01312857	II	Resected Colorectal Cancer Liver Metastasis with Wild Type RAS	panitumumab	Active, not recruiting
NCT03732235	NA	Refractory Colorectal Cancer Liver Metastases	TACE+ systemic Bevacizumab; FOLFIRI+Bevacizumab; TACE	Recruiting
NCT04126655	I/II	Colorectal Cancer Liver Metastases	Arfolitixorin + 5-FU; Calciumfolinate + 5-FU	Recruiting
NCT03477019	I/II	Liver Metastasis from Breast- and Colorectal Cancer	SonoVue; Focused Ultrasound	Recruiting
NCT03458975	II	Colorectal Cancer Liver Metastases	Contrast enhanced ultrasound; Sonoporation	Recruiting
NCT04021277	I	Colorectal Cancer Liver Metastases	PS101-mediated Acoustic Cluster Therapy	Recruiting
NCT03493048	II	RAS wildtype Unresectable Colorectal Cancer Liver Metastases	Irinotecan; Cetuximab; 5-fluorouracil; Oxaliplatin; Leucovorin	Recruiting
NCT03801915	II	Colorectal Cancer Liver Metastases	MVT-5873	Recruiting
NCT02172651	Early Phase 1	Stage I-III Colon Cancer or Resectable Colon Cancer Liver Metastases	Vitamin D3	Recruiting
**Ablation, radiotherapy, surgery**				
NCT03088150	NAII	Colorectal Cancer Liver Metastases	Thermal ablation; Surgical resection	Recruiting
NCT03963726/ NCT03654131/ NCT04081168	NA/II	Colorectal Cancer Liver Metastases	stereotactic radiotherapy; microwave ablation	Recruiting
NCT02185443	II	Unresectable Colorectal Cancer Liver Metastases	SBRT	Recruiting
NCT04491929	NA	Refractory Colorectal Cancer Liver Metastases	Selective Internal Radiation Therapy With 90Y Resin	Recruiting
NCT03895723	NA	Colorectal Cancer Liver Metastases	laparoscopic and robotic liver resection or open surgery	Recruiting
NCT02954913	NA	Colorectal Cancer Liver Metastases	Simultaneous Resection	Recruiting
NCT03803436	II	Unresectable Colorectal Cancer Liver Metastases	liver transplantation vs triplet chemotherapy+antiEGFR	Recruiting
NCT02864485/ NCT01479608/ NCT02597348	NA; II; III	Unresectable Colorectal Cancer Liver Metastases	liver transplantation	Recruiting
NCT02215889	I/II	Colorectal Cancer Liver Metastases	Partial Liver Segment 2/3 Transplantation	Recruiting
NCT03494946	NA	Colorectal Cancer Liver Metastases	Liver transplantation vs Chemotherapy	Recruiting
NCT04161092	NA	non-resectable/ non-abatable colorectal liver metastases	Liver transplantation Ltx or best alternative care	Not yet recruiting
NCT03488953	NA	Isolated, Irresectable Colorectal Liver Metastases	Living donor liver transplantation with two-staged hepatectomy	Recruiting
NCT03577665	NA	Colorectal Cancer Liver Metastases	Curative Proton Beam Therapy	Recruiting
NCT04108481	I/II	Colorectal Cancer Liver Metastases	Durvalumab; Yttrium-90 RadioEmbolization	Recruiting
**Immunotherapy**				
NCT02754856	I	Resectable Colorectal Cancer Liver Metastases	Durvalumab; Tremelimumab	Recruiting
NCT03370198	I	Unresectable Liver Metastases from Colorectal Cancer	Hepatic Transarterial Administrations of NKR-2(modified T cells)	Active, not recruiting
NCT02850536	I	Liver Metastases or Pancreas Cancer	anti-CEA CAR-T cells	Active, not recruiting
NCT04513431	Early Phase 1	Stage III Colorectal Cancer Colorectal Cancer Liver Metastasis	Anti-CEA-CAR T	Not yet recruiting
**Combination therapy**				
NCT04062721	Ib/II	Unresectable Colorectal Liver Metastases	radiofrequency ablation (RFA) plus in situ TLR agonist and GM-CSF	Not yet recruiting
NCT04202978	I/II	Colorectal Cancer Liver Metastases	Camrelizumab Combined With Apatinib XELOX RFA	Recruiting
NCT03223779	I/II	Colorectal Cancer Liver Metastases	TAS-102; Photon SBRT	Recruiting
NCT02738606	II	Resectable Colorectal Cancer Liver Metastases and unresectable Colorectal Cancer Lung Metastases	liver surgery and chemotherapy	Recruiting
NCT03127072	IV	Unresectable Colorectal Cancer Liver Metastases	Radiofrequency Ablation (RFA); chemotherapy ± target therapy	Recruiting
NCT04562727	NA/II	Colorectal Cancer Liver Metastases	Microwave Ablation Combined with Chemotherapy	Not yet recruiting
NCT03135652	II	Colorectal Cancer Liver Metastases Receiving Surgery or Radiofrequency Ablation	Adjuvant SBRT; Chemotherapy	Recruiting
NCT03101475	II	Colorectal Cancer Liver Metastases	Durvalumab (MEDI4736); Tremelimumab; Sterotactic body radiation therapy (SBRT); Radiofrequency ablation (RFA)	Recruiting
NCT04508140	II	Colorectal or Gastric/GEJ Cancer with Liver Metastasis	BO-112 with Pembrolizumab	Recruiting
NCT03507699	I	Colorectal Cancer Liver Metastases	Liver radiation therapy; Nivolumab Injection; Ipilimumab Injection; CMP-001	Recruiting
NCT04166383	II	Colorectal Cancer Liver Metastases	VB-111; Nivolumab	Recruiting
NCT03785210	II	Refractory Primary Hepatocellular Carcinoma or Liver Dominant Metastatic Cancer from Colorectal or Pancreatic Cancers	nivolumab; tadalafil; oral vancomycin	Recruiting
NCT04430985	II	Colorectal Cancer Liver Metastases	Oxaliplatin; 5-Fluorouracil; Leucovorin; Nivolumab; Ipilimumab	Recruiting
NCT03698461	II	Colorectal Cancer Liver Metastases	Atezolizumab; Bevacizumab; Oxaliplatin; Levoleucovorin; 5-FU	Recruiting
NCT03310008	I	Colorectal Cancer with Potentially Resectable Liver Metastases	NKR-2(modified T cells) with FOLFOX	Active, not recruiting
NCT04046445	I/II	Stage IV Colorectal Cancer	ATP128; BI 754091	Recruiting

NA, not applicable.

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
