# Peer review of "Liver Immune Microenvironment and Metastasis from Colorectal Cancer-Pathogenesis and Therapeutic Perspectives"

_cancers, 2021, doi:10.3390/cancers13102418_

Round 1
Reviewer 1 Report
This is a comprehensive review paper on the immune microenvironment and therapeutic perspectives of colorectal cancer liver metastasis. I greatly enjoyed the manuscript which is very well written and concise.
A couple of comments:
- Since histological growth patterns play a critical role in colorectal cancer liver metastasis, I suggest to include a section that explains the histological growth patterns and their clinical importance.
- The impact of immune cells in establishing non-angiogenic metastatic lesions and resistance to anti-angiogenic agents in colorectal cancer liver metastasis should be discussed.
- Some abbreviations are missing the full term e.g. PDAC, BMP.
- In section 2.1.2., the function of matrix metalloproteinase (MMP) should be further explained.
Author Response
Reviewer 1:
This is a comprehensive review paper on the immune microenvironment and therapeutic perspectives of colorectal cancer liver metastasis. I greatly enjoyed the manuscript which is very well written and concise.
Response to Reviewer 1:
We appreciate the reviewer’s compliment on our ‘comprehensive review’, which is ‘well written and concise’.
Major comments:
1. Since histological growth patterns play a critical role in colorectal cancer liver metastasis, I suggest to include a section that explains the histological growth patterns and their clinical importance.
Response:
We thank the reviewer for this suggestion. Histological growth patterns are indeed of great clinical significance and associated with patient prognosis. Thus, we have added an additional paragraph in the manuscript accordingly the revised section 1.2 to describe the difference of histological growth patterns of CRC liver metastasis and their preference of immune infiltration accordingly.
2. The impact of immune cells in establishing non-angiogenic metastatic lesions and resistance to anti-angiogenic agents in colorectal cancer liver metastasis should be discussed.
Response:
We agree with the reviewer’s comment. The impact of immune cells, such as neutrophils recruitment was associated with anti-angiogenic drug resistance in CRC liver metastases with replacement histological growth pattern, in which this discussion is now added in the revised section 2.3.1.
3.Some abbreviations are missing the full term e.g. PDAC, BMP.
Response:
We have now double-checked the manuscript, and all the full terms have been added.
4. In section 2.1.2., the function of matrix metalloproteinase (MMP) should be further explained.
Response:
A brief explanation of MMP function in cancer progression and metastasis accordingly has now been included in the revised section 2.1.2.

Reviewer 2 Report
The review carried out by the authors aimed to summarize the immune microenvironment in the metastasic liver of colorectal cancer, with the pathogenesis and therapeutic alternatives. The subject of the review is very interesting and can provide very valuable information for further deepen into the mechanisms involved in the liver metastasis of colorectal cancer and its possible treatments. However there are some questions which the authors have to consider.
The title “Liver Immune Microenvironment and Metastasis from Colorec-tal Cancer-Pathogenesis and Therapeutic Perspectives” is not totally correct. The title indicates that the review focuses on the inmune population of the liver and its implication on liver metastasis of colorectal cancer. However, the review summarizes the implication of different cell types and ECM on liver metastasis and short immune information. The authors should deepen into the 2.3 point with more information about different immune cells (for example NK cells are very important regulating the liver environment) and molecules such as PD-1, PDL-1, CTLA4, among others, which are important to introduce immunotherapy. Moreover, the relation of non parenquimal or parenquimal cells with an immune cells and its implication on metastasis is not described and it should be an important aim according the title.- The figures 1 and 2 are very informative, however, there is difficult to continue all the information of the figures with the text. The authors could divide the figures 1 and 2 in more figures and relate better with the text, which allows a better understanding of the concepts.
- The authors report that tumor activated hepatocyte and myofibroblast could affect the phenotype of primary CRC cell by upregulating liver metastasis gene expression (including liver-specific proinflammatory, immunoregulatory gene, etc.) [25]. In this case, although there is a reference, in order to make more complete and understable the review, the information of the genes which they are reporting is very general, they should mention the genes and amplify the information.
- The authors should revise and correct the abreviatures
Author Response
Reviewer 2:
The review carried out by the authors aimed to summarize the immune microenvironment in the metastasic liver of colorectal cancer, with the pathogenesis and therapeutic alternatives. The subject of the review is very interesting and can provide very valuable information for further deepen into the mechanisms involved in the liver metastasis of colorectal cancer and its possible treatments. However, there are some questions which the authors have to consider.
Response to Reviewer 2:
We appreciate the reviewer’s compliment on our review with ‘interesting’ and ‘valuable’.
1. The title “Liver Immune Microenvironment and Metastasis from Colorectal Cancer-Pathogenesis and Therapeutic Perspectives” is not totally correct. The title indicates that the review focuses on the immune population of the liver and its implication on liver metastasis of colorectal cancer. However, the review summarizes the implication of different cell types and ECM on liver metastasis and short immune information. The authors should deepen into the 2.3 point with more information about different immune cells (for example NK cells are very important regulating the liver environment) and molecules such as PD-1, PDL-1, CTLA4, among others, which are important to introduce immunotherapy. Moreover, the relation of non parenquimal or parenquimal cells with an immune cell and its implication on metastasis is not described and it should be an important aim according the title.
Response:
We thank the reviewer for this good suggestion. We have highlighted the immunoregulatory roles of liver parenchymal/non-parenchymal cells in controlling CRC liver metastasis, including the examples of hepatocyte derived cytokines and chemokines in promoting MDSC expansion and liver recruitment on CRC liver metastasis (section 2.1.2), as well as Kupffer cell in suppressing anti-tumor T cell response via IL-10 and PD-L1 (section 2.2.3). To further emphasize the liver immune microenvironment, we now have revised our review accordingly by including the following information:
a) Summary of the potential crosstalk between liver parenchymal/non-parenchymal cells and different immune cells in regulating CRC liver metastasis, including the crosstalk of hepatocyte-MDSC-NKT cell, HSC-monocyte in the revised section 2.1.2 and 2.2.1.
b) The roles of NK and NKT cells in CRC liver metastasis as a new section 2.3.3 and 2.3.4.
c) Immune checkpoint molecules in CRC liver metastasis as a new section 2.5.
2. The figures 1 and 2 are very informative, however, there is difficult to continue all the information of the figures with the text. The authors could divide the figures 1 and 2 in more figures and relate better with the text, which allows a better understanding of the concepts.
Response:
We thank the reviewer for the suggestion. Since the description on the liver immune microenvironment shaped by different hepatocyte-derived factors and tumor cells is informative as the reviewer mentioned, we think figures including all the information would be meaningful for understanding the sequential events of metastasis. To this end, we revised our description of these two figures and highlighted the location of indicated cells in the main text accordingly, including ‘Figure 2, in sinusoid/space of disse/parenchyma” in the revised sections 2.2.1, 2.2.2, 2.2.3, 2.3.1, 2.3.2.
3. The authors report that tumor activated hepatocyte and myofibroblast could affect the phenotype of primary CRC cell by upregulating liver metastasis gene expression (including liver-specific proinflammatory, immunoregulatory gene, etc.) [25]. In this case, although there is a reference, in order to make more complete and understable the review, the information of the genes which they are reporting is very general, they should mention the genes and amplify the information.
Response:
We thank the reviewer for the suggestion. We have included examples of the related genes in revised section 2.1.1, and further expanded the description on the roles of TGF-beta in promoting CRC metastasis.
4. The authors should revise and correct the abbreviates
Response:
We have double checked the manuscript, revised and corrected them accordingly.

Reviewer 3 Report
This a comprehensive review for CRC liver metastasis. I have few comments for authors-
1.What is the new insight in this review regarding CRC liver metastasis from already published CRC liver metastasis reviews.
2. Liver immune microenvironment is very suitable for any metastasis, how it is different for CRC and specifically CRC metastasis. Is CRC metastasis of liver follow the similar mechanism as other cancers.
Author Response
Reviewer 3:
This a comprehensive review for CRC liver metastasis. I have few comments for authors-
Response to Reviewer 3:
We appreciate the reviewer’s compliment on our ‘comprehensive’ review. The comments provide new insight and good suggestions.
1. What is the new insight in this review regarding CRC liver metastasis from already published CRC liver metastasis reviews.
Response:
Although a number of reviews has highlighted certain important features of CRC liver metastasis, our review has emphasized the uniqueness of liver immune microenvironment in facilitating CRC liver metastasis. In particular, we have added some new findings such as hepatocyte-intrinsic CCRK signaling in promoting liver immunosuppressive MDSC that facilitates metastatic CRC outgrowth, and summarized the current clinical trials and new therapeutic perspectives. Immunotherapies such as immune checkpoint inhibitors, cancer vaccines and CAR T cell therapy have been investigated in treating CRC liver metastasis patients with limited efficacy owing to the inhibitory impact of the tumor immune microenvironment. Based on these new findings, we have provided new therapeutic insight of targeting the liver immune microenvironment e.g. CDKs that drives premetastatic niche formation and immunosuppressive environment in liver (revised section 3.2.1). We believe these new insight is important in understanding the mechanisms and therapeutic advancement.
2. Liver immune microenvironment is very suitable for any metastasis, how it is different for CRC and specifically CRC metastasis. Is CRC metastasis of liver follow the similar mechanism as other cancers?
Response:
We thank the reviewer for pointing out this interesting aspect. We choose CRC liver metastasis as our focus because CRC is the most common cancer that shows a preference to metastasize to liver when compared to other cancer types. The anatomic proximity between liver and colon may partially explain this clinical phenomenon that has been already summarized by other reviews. In addition, specific signals derived from CRC cells are different from other cancers, which may also impact on CRC liver metastasis. On the other hand, we and others have further highlighted the importance of liver immunosuppressive microenvironment in promoting CRC metastasis, in which features may be shared by other primary cancers derived liver metastasis. Nevertheless, the mechanisms underlying liver tropism in different cancers remain unclear, which is very important to be further explored. We have now added this discussion in the revised section 4.

Reviewer 4 Report
The review contains a great deal of information and the sections have been done very well. However there are some omissions , which I have indicated below and there needs to be more clarity on when the authors are referring to metastatic niche vs tumor establishment and growth. The title is misleading. The review looks at the microenvironment and does not focus on the immune microenvironment. There are some key papers on the immune environment that should be included and addressed. Also, since the authors wish to address the seeding of tumor cells there should be mention of circulating tumor cells and studies demonstrating the role of single cells vs clumps in preparing the metastatic niche.
Specific comments:
Throughout the text the word metastasis and metastases are used incorrectly. Please review and correct were the authors refer to the singular as metastasis and the plural metastases. For example in this sentence at the end of section 1.3 should be metastases. This needs to be reviewed and corrected in the entire
In section 2, reference 19 is on pancreatic cancer and this section refers to CRC only. If the authors want to include the reference, they must indicate that some of the data comes from other primary cancers. Otherwise this is misleading. Therefore, in the first sentence the authors should indicate that the "substantial amount of studies" are not only from CRC but other primary cancers. Furthermore, they should also address any differences between different primary cancers and how the environment may be differentially prepared depending on the cancer type. This should be addressed as the paper focus is on CRC and a number of references are from different primaries.
In section 1.2, last paragraph a list of references of current clinical trials should be included.
In section 1.3 the sentence: "least known aspect" what does this mean?
Should be: "remains as one of the poorly characterized mechanisms "
The term CLM should be defined.
Section 2.2.1: Since there is a larger section on CAFs this heading3. should include "....and Cancer associated fibroblasts (CAF)"
Section 2.3.1: Should also include study demonstrating that neutrophils can also express ECM remodeling proteins such as lysloxidases (lox) and therefore their role may be multifunctional. (Palmieri, V. et al. Neutrophils expressing lysyl oxidase–like 4 protein are present in colorectal cancer Therapy, metastases resistant to anti-angiogenic. J. Pathol. (2020) doi:10.1002/path.5449.).
Section 2.3.2 In this study it is important to understand seeding vs growth. The number of mets and size of mets were reduced therefore the MAMs were involved in not only seeding but also growth. The authors should include a statement in their conclusion that a number of studies using animal models don't always differentiate between seeding and growth. Also, in human samples this remains to be validated.
Section 2.3.3 “ In a retrospective study, high Treg infiltration predicted poor clinical outcome of CRC liver metastasis patients, indicating the pro-metastatic role of Treg in liver metastasis [71]. “ This is in established tumors and therefore does not confirm the role in pre-metastatic niche. The authors are including studies that look at established metastases in human samples to "seeding" of cancer cells. This is confusing and should be explained better.
Section 3.2: This section is well described however the relation to metastatic niche is not clear and in some of the examples this may have nothing to do with niche formation. Especially studies in humans where the tumors are already established. This affects the growth once seeded and should be indicated as the step after seeding. Clarification is needed in their statements.
3.3: The examples in this section are in relation to lung mets for the mechanism. As described earlier by the authors that the liver environment is different this section should be omitted as a thorough review of the literature in CRC is not shown.
Section 4 There are other strategies such as staged resections and PVE etc. The authors should refer to other reviews as theirs is not inclusive of all studies and only demonstrates a few studies. Also the authors should mention that MSI high patients can be treated with immunotherapies and are very successful.
Section 4.1.1: “Patients with good liver function and general condition and without metastasis in other organs except liver, are suitable for surgical resection.” This is not an accurate statement. Patients with small lung nodules will still be resected as the lung mets can be ablated in most cases.
Section 4.1.3: PVE is also used in staged resections with patients with bilateral disease. The authors completely omitted this treatment.
Section 4.2.2: The authors are confusing pre and pro metastatic niche with target of treatment. All treatments for liver mets are on established lesions and therefore there is no evidence on the effect on the metastatic niche. Adjuvant therapy for the primary has been shown to reduce liver mets and the authors should include this as they are focused on the premetastatic niche. The review is moving from pre-metastatic niche to established mets. This needs to be clearly outlined.
Section 4: The major challenge is not only recurrence, but a larger portion of patients are unresectable! There are a number of studies on patient stratification based on histopathological growth patterns that have been omitted. There is now a growing body of evidence that not all mets are the same and this affects the overall survival of patients. Also, there are a number of publications demonstrating the immune landscape of these mets are different. There is a body of data and publications that have been omitted and overall the clinical section appears weak and should be updated with recent reviews, some of which have been published in Cancers.
Author Response
Reviewer 4:
The review contains a great deal of information and the sections have been done very well. However, there are some omissions, which I have indicated below and there needs to be more clarity on when the authors are referring to metastatic niche vs tumor establishment and growth. The title is misleading. The review looks at the microenvironment and does not focus on the immune microenvironment. There are some key papers on the immune environment that should be included and addressed. Also, since the authors wish to address the seeding of tumor cells there should be mention of circulating tumor cells and studies demonstrating the role of single cells vs clumps in preparing the metastatic niche.
Response to Reviewer 4:
We appreciate the reviewer’s compliment that ‘The review contains a great deal of information and the sections have been done very well.’ As suggested, we have revised our review accordingly.
Specific comments:
- Throughout the text the word metastasis and metastases are used incorrectly. Please review and correct were the authors refer to the singular as metastasis and the plural metastases. For example in this sentence at the end of section 1.3 should be metastases. This needs to be reviewed and corrected in the entire
Response:
We have reviewed the manuscript and corrected them accordingly.
- In section 2, reference 19 is on pancreatic cancer and this section refers to CRC only. If the authors want to include the reference, they must indicate that some of the data comes from other primary cancers. Otherwise this is misleading. Therefore, in the first sentence the authors should indicate that the "substantial amount of studies" are not only from CRC but other primary cancers. Furthermore, they should also address any differences between different primary cancers and how the environment may be differentially prepared depending on the cancer type. This should be addressed as the paper focus is on CRC and a number of references are from different primaries.
Response:
We thank the reviewer for this suggestion. In the first sentence of section 2.1.1, we want to emphasize the roles of hepatocyte-derived cytokines and chemokines in driving pre-metastatic niche formation and supporting liver metastasis. Thus, we used the new interesting finding in ‘early pancreatic cancer development’ that besides tumor-derived factors, hepatocyte-derived factors also play pivotal roles in pre-metastatic niche formation, in which features may be shared by CRC. We have also revised this section by adding a new reference (ref 34) from CRC to address this point.
- In section 1.2, last paragraph a list of references of current clinical trials should be included.
Response:
We have included the references (ref 21-24) accordingly.
- In section 1.3 the sentence: "least known aspect" what does this mean? Should be: "remains as one of the poorly characterized mechanisms "
Response:
We have revised the sentence to ‘remains one of the poorly characterized aspects’ in section 1.3 accordingly.
- The term CLM should be defined.
Response:
We have changed ‘CLM’ into ‘CRC liver metastasis’ in legend of Figure 1 accordingly.
- Section 2.2.1: Since there is a larger section on CAFs this heading3. should include "....and Cancer associated fibroblasts (CAF)"
Response:
We thank the reviewer for the comment, and the subtitle of section 2.2.1 has been revised to ‘Hepatic stellate cell (HSC) and cancer associated fibroblasts (CAF)’ accordingly.
- Section 2.3.1: Should also include study demonstrating that neutrophils can also express ECM remodeling proteins such as lysloxidases (lox) and therefore their role may be multifunctional. (Palmieri, V. et al.Neutrophils expressing lysyl oxidase–like 4 protein are present in colorectal cancer Therapy, metastases resistant to anti-angiogenic. Pathol. (2020) doi:10.1002/path.5449.).
Response:
We thank the reviewer for the suggestion. We have now added this new reference (ref 76) and related information in the revised section 2.3.1.
- Section 2.3.2 In this study it is important to understand seeding vs growth. The number of mets and size of mets were reduced therefore the MAMs were involved in not only seeding but also growth. The authors should include a statement in their conclusion that a number of studies using animal models don't always differentiate between seeding and growth. Also, in human samples this remains to be validated.
Response:
We agree with the reviewer’s comment. Indeed, it is difficult to differentiate the effect of inhibiting tumor cell seeding or growth in most studies using animal models, when reduced liver metastasis was observed. We have included this discussion in revised section 2.3.2.
- Section 2.3.3 “In a retrospective study, high Treg infiltration predicted poor clinical outcome of CRC liver metastasis patients, indicating the pro-metastatic role of Treg in liver metastasis [71]. “ This is in established tumors and therefore does not confirm the role in pre-metastatic niche. The authors are including studies that look at established metastases in human samples to "seeding" of cancer cells. This is confusing and should be explained better.
Response:
We agree with the reviewer’s comment. Indeed, this reference indicated the role of Treg in established CRC liver metastases. We have now revised in section 2.3.5 (original section 2.3.3) as ‘ infiltrating Treg cells support the growth of established CRC liver metastases’ accordingly.
- Section 3.2: This section is well described however the relation to metastatic niche is not clear and in some of the examples this may have nothing to do with niche formation. Especially studies in humans where the tumors are already established. This affects the growth once seeded and should be indicated as the step after seeding. Clarification is needed in their statements.
Response:
We thank the reviewer for the suggestion. Since the description of tumor cell seeding and colonization in CRC metastasis is well clarified by the current understanding as mentioned by the reviewer, we now focused on emphasizing the role of ECM in supporting the growth of metastasized tumor cells in the new section 2.4 and delete the original section 3.
- 3: The examples in this section are in relation to lung mets for the mechanism. As described earlier by the authors that the liver environment is different this section should be omitted as a thorough review of the literature in CRC is not shown.
Response:
We agree with the reviewer’s comment. Since the importance of dormant CTC reaction in CRC liver metastasis is not supported by direct evidence as pointed by the reviewer, we have deleted this section accordingly.
- Section 4 There are other strategies such as staged resections and PVE etc. The authors should refer to other reviews as theirs is not inclusive of all studies and only demonstrates a few studies. Also the authors should mention that MSI high patients can be treated with immunotherapies and are very successful.
Response:
We thank the reviewer for the suggestion. We have now included new references (ref 117) with a brief introduction in the revised section 3.1.1 (original section 4). The efficacy of immunotherapies in treating MSI high patients was included and further explored in the revised section 3.2.2 (original section 4).
- Section 4.1.1: “Patients with good liver function and general condition and without metastasis in other organs except liver, are suitable for surgical resection.” This is not an accurate statement. Patients with small lung nodules will still be resected as the lung mets can be ablated in most cases.
Response:
We thank the reviewer for the comment. As the reviewer pointed, patients with good liver function and general condition and without metastasis in other organs except liver, are also suitable for surgical resection. In particular cases, patients with metastases from other organs may also be resectable. In addition, radiofrequency ablation (RFA) is widely performed for the treatment of CRC lung metastases. We have now included this discussion in the revised section 3.1.1 (original section 4).
- Section 4.1.3: PVE is also used in staged resections with patients with bilateral disease. The authors completely omitted this treatment.
Response:
We have briefly mentioned PVE in the original section 4.1.3 (now section 3.1.3). As PVE is less frequently used in treating CRC liver metastasis compared to surgical resection and ablation, we did not explore the PVE in details due to the word limitation.
- Section 4.2.2: The authors are confusing pre and pro metastatic niche with target of treatment. All treatments for liver mets are on established lesions and therefore there is no evidence on the effect on the metastatic niche. Adjuvant therapy for the primary has been shown to reduce liver mets and the authors should include this as they are focused on the premetastatic niche. The review is moving from pre-metastatic niche to established mets. This needs to be clearly outlined.
Response:
We thank the reviewer for the suggestion. We have now further clarified this point in the revised section 3.2.2 accordingly (original section 4.2.2).
- Section 4: The major challenge is not only recurrence, but a larger portion of patients are unresectable! There are a number of studies on patient stratification based on histopathological growth patterns that have been omitted. There is now a growing body of evidence that not all mets are the same and this affects the overall survival of patients. Also, there are a number of publications demonstrating the immune landscape of these mets are different. There is a body of data and publications that have been omitted and overall the clinical section appears weak and should be updated with recent reviews, some of which have been published in Cancers.
Response:
We thank the reviewer for the comment. Histological growth patterns are indeed of great clinical significance and associated with patient prognosis. Thus, we have added an additional paragraph and new references (ref 11-18) in the manuscript accordingly the revised section 1.2 to describe the difference of histological growth patterns of CRC liver metastasis and their preference of immune infiltration accordingly.

Round 2
Reviewer 2 Report
The authors have included all the sugestions and improved the article.